# Emerging Immunotherapy for Acute Myeloid Leukemia

**DOI:** 10.3390/ijms22041944

**Published:** 2021-02-16

**Authors:** Rikako Tabata, SungGi Chi, Junichiro Yuda, Yosuke Minami

**Affiliations:** 1Department of Hematology, National Cancer Center Hospital East, Kashiwa 277-8577, Japan; tabata.rikako@kameda.jp (R.T.); schi@east.ncc.go.jp (S.C.); jyuda@east.ncc.go.jp (J.Y.); 2Department of Hematology, Kameda Medical Center, Kamogawa 296-8602, Japan

**Keywords:** acute myeloid leukemia (AML), immune check-point inhibitor (ICI), bispecific T-cell engager (BiTE), dual-affinity retargeting (DART), trispecific killer cell engager (TriKE), chimeric antigen receptor (CAR)

## Abstract

Several immune checkpoint molecules and immune targets in leukemic cells have been investigated. Recent studies have suggested the potential clinical benefits of immuno-oncology (IO) therapy against acute myeloid leukemia (AML), especially targeting CD33, CD123, and CLL-1, as well as immune checkpoint inhibitors (e.g., anti-PD (programmed cell death)-1 and anti-CTLA4 (cytotoxic T-lymphocyte-associated protein 4) antibodies) with or without conventional chemotherapy. Early-phase clinical trials of chimeric antigen receptor (CAR)-T or natural killer (NK) cells for relapsed/refractory AML showed complete remission (CR) or marked reduction of marrow blasts in a few enrolled patients. Bi-/tri-specific antibodies (e.g., bispecific T-cell engager (BiTE) and dual-affinity retargeting (DART)) exhibited 11–67% CR rates with 13–78% risk of cytokine-releasing syndrome (CRS). Conventional chemotherapy in combination with anti-PD-1/anti-CTLA4 antibody for relapsed/refractory AML showed 10–36% CR rates with 7–24 month-long median survival. The current advantages of IO therapy in the field of AML are summarized herein. However, although cancer vaccination should be included in the concept of IO therapy, it is not mentioned in this review because of the paucity of relevant evidence.

## 1. Introduction

Since the Food and Drug Administration (FDA) approved ipilimumab, the first-in-class anti-CTLA-4 (cytotoxic T-lymphocyte-associated protein 4) antibody, for melanoma in March 2011 [1], immune checkpoint inhibitors (ICIs) such as nivolumab, pembrolizumab, atezolizumab, and durvalumab have been eagerly developed and are now practically available for a variety of malignant tumors. For the majority of solid tumors, immuno-oncology (IO) therapy has been involved in mainstream cancer treatment along with molecular targeted therapy, conventional chemotherapy, and radiation therapy. In the field of hematologic malignancies, the first bispecific T-cell engager (BiTE) blinatumomab, a CD19- and CD3-targeted bispecific antibody, achieved complete remission with 6.7 month-long duration in 32% of the patients with B-cell lymphoblastic leukemia (B-ALL) in a clinical study [2] and has shown efficacy in non-Hodgkin lymphomas (NHLs) [3,4]. Currently, chimeric antigen receptor (CAR)-T therapy also accounts for an essential part of IO therapy against hematologic malignancies. The first-in-class CD19-targeted CAR-T tisagenlecleucel (also known as tisa-cel) was approved by the FDA for relapsed/refractory B-ALL in August 2017 and diffuse large B cell lymphoma (DLBCL) in May 2018, with overall remission rates of 83% and 52%, respectively [5,6]. A multicenter phase 1/2 study ZUMA-1 also showed somwhat better response rates (82%) of another CD19-targeted CAR-T axicabtagene ciloleucel (also known as axi-cel) for refractory DLBCL [7]. Traditional monoclonal antibodies (e.g., rituximab, alemtuzumab, obinutuzumab, and mogamulizumab) and antibody-drug conjugates (ADCs) (e.g., gemtuzumab ozogamicin (GO) and brentuximab vedotin) are also used in IO therapy. Novel agents of CAR-Ts, ADCs, and bispecific antibodies are currently under development.

Intensive chemotherapy (e.g., anthracyclines and cytarabine) with or without hematopoietic stem cell transplantation (SCT) has been the mainstay of the curative treatment of acute myeloid leukemia (AML), presenting a dismal prognosis for relapsed/refractory or intolerable cases which mostly require palliative care. For CD33-positive AML, GO showed an overall response rate of 63% in relapsed/refractory cases [8], along with survival benefit in newly diagnosed cases when combined with conventional chemotherapy [9]. Due to the paucity of favorable immune targets in leukemic cells, IO approaches for AML have been limited. However, as immune inhibitory molecules and cancer-related antigens on leukemic cells have been discovered, novel IO drugs have been recently developed in the field of AML and are expected to become another treatment option. Recent representative advantages in terms of IO therapy are summarized in this review.

## 2. Molecules Involved in IO Therapy

### 2.1. Immune Checkpoint Molecules

#### 2.1.1. Programmed Cell Death 1 (PD-1), Programmed Cell Death-Ligand 1 (PD-L1), and Cytotoxic T-Lymphocyte-Associated Protein 4 (CTLA-4)

The inhibitory surface receptor programmed cell death 1 (PD-1) (UniProtKB-Q15116) on activated T cells is encoded by the *PDCD1* gene on chromosome 2q37.3. Along with its ligands PD-L1 and PD-L2, also known as CD274 and CD273, PD-1 plays an important role in maintaining self-tolerance [10] and is often involved in immune escape in cancer by inhibiting the direct cytotoxic activity of effector CD8-positive T cells on tumor cells [11]. CTLA-4 on activated T cells, which is encoded by the CTLA4 gene on chromosome 2q33.2, also has a crucial role in attenuating T cell activation in peripheral lymph nodes by preventing CD28 on T cells to bind its co-stimulatory counterparts B7 family ligands (CD80 and CD86) on antigen-presenting cells [12,13]. An in vivo study of murine myelogenous leukemia suggested that blockade of B7-1 (CD80) and not B7-2 (CD86) by CTLA-4 contributed to the attenuation of anti-leukemic immunity [14].

An observational study at the MD Anderson Cancer Center analyzed bone marrow and peripheral blood specimens from 124 patients with myelodysplastic syndrome (MDS), chronic myelo-monocytic leukemia (CMML), and AML who received hypomethylating agents (HMAs) and reported that PD-1 and PD-L1 expression on CD34-positive cells were found in 7% and 20% of the patients, respectively [15]. In 57% of previously untreated patients, PD-L1 and PD-L2 expression on peripheral blood mononuclear cells (PBMNCs) increased more than twice during the first cycle of HMA. These patients had a shorter median survival than those who did not (4.7–6.6 vs. 11.7–12.5 months), suggesting the negative impact of PD-L1 and PD-L2 on the anti-tumor effect of HMAs. Upregulation of CTLA-4 on PBMNCs was also observed in 8% of the patients. Another study suggested that PD-L1 expression was higher in relapsed cases and associated with poor prognosis [16]. Epigenetic analysis of 197 AML specimens revealed that the less methylated promoters of PD-L1 and PD-L2 gene in leukemic cells were an independent negative prognostic factor [17]. Analysis of bone marrow samples from nine refractory/relapsed AML patients showed a higher proportion (22%) of CD8-positive T cells co-expressing PD-1 and larger T-cell clonal expansion measured by T-cell receptor rearrangement compared with healthy donor samples [18]. PD-1 and OX40 on bone marrow T cells were more frequently found in relapsed AML samples than in newly diagnosed ones [19]. A report from China showed that PD-1 expression was seen in 33.8% of the peripheral CD3-positive lymphocytes in patients with previously untreated de novo AML and was correlated with the increased expression of exhaustion markers such as CD244 and CD57 [20]. However, other experiments suggested that PD-1 expression does not result in functional impairment of T cells, but rather correlates with a shift to memory cells [21]. Twenty-three samples from patients with AML were compared with those of 30 healthy controls. Although relatively high (>30%) PD-1 expression on CD8-positive T cells was observed in 3 of 23 (13%) AML samples, the median percentages did not differ significantly compared with healthy controls (median 15.6%). Other immune inhibitory markers, CD244, CD160, and TIM-3, were also not significantly expressed. Instead, PD-1 was upregulated in peripheral blood specimens of patients with AML who relapsed after either intensive chemotherapy or allogeneic stem cell transplantation (allo-SCT) compared with those of the same patients at the time of diagnosis.

#### 2.1.2. T-Cell Immunoglobulin and Mucin-Domain Containing-3 (TIM-3)

The cell surface receptor T-cell immunoglobulin and mucin-domain containing-3 (TIM-3), also known as hepatitis A virus cellular receptor 2 (HAVcr-2), is encoded by the HAVCR2 gene on chromosome 5q33.3. TIM-3 is normally expressed on T-helper type 1 (Th1) lymphocytes, regulatory T cells (Treg), and natural killer (NK) cells. TIM-3 regulates macrophage activation [22], promotes immunological tolerance by inhibiting Th1-mediated responses [23], attenuates T-cell receptor (TCR)-induced signaling in CD8-positive T cells [24], and inhibits Th17 responses when expressed on Tregs [25]. The first identified ligand for TIM-3 is galectin-9 [26], which is also a ligand for P4HB and CD44 [27]. It contributes to the stabilization/empowerment of induced Tregs [27], helps mesenchymal stromal cells in suppressing T cells [28], and inhibits NK cell activity [29]. TIM-3 is also able to bind phosphatidylserine (PtdSer) [30], HMGB1 [31], and CEACAM1 [32] to prevent the activation of the immune response.

TIM-3 was found in approximately 6% of the bone marrow T cells in newly diagnosed AML patients, and the proportion was larger (11.5–18.5%) in FLT3-ITD mutated cases [33]. Simultaneous expression of PD-1 and TIM-3 on peripheral T cells was associated with AML relapse after allo-SCT [34]. An in vitro experiment using murine AML cells demonstrated that co-expression of TIM-3 and PD-1 on CD8-positive T cells was enhanced during disease progression and inhibition of either molecule alone did not attenuate tumor activity [35]. TIM-3 is also expressed on NK cells [36], while galectin-9 is found in AML blasts [37]. TIM-3/galectin-9 interaction leads to the production of IFN-gamma by NK cells, resulting in indoleamine 2,3-dioxygenase 1 (IDO1) expression in AML cells [38]. IDO1-positive AML cells gained the ability to negatively regulate NK cell degranulation, which can be responsible for the immune escape of leukemic cells. Another in vitro study suggested that AML cells protect themselves by producing soluble TIM-3 to form a TIM-3-galectin-9 complex that attenuates NK cell-mediated cytotoxicity [39].

TIM-3 inhibitors are currently under early phase evaluation as monotherapies or in combination with PD-1/PD-L1 inhibitors for patients with advanced tumors. Preclinically, CAR-T therapy bispecific for CD13 and TIM-3 has been efficiently screened by a nanobody-based technology and shows eradication of leukemic cells in mouse models [40].

#### 2.1.3. Lymphocyte Activation Gene-3 Protein (LAG-3)

The inhibitory receptor lymphocyte activation gene-3 protein (LAG-3), also known as CD223, is encoded by the LAG-3 gene on chromosome 12p13.31. LAG-3 is normally expressed on activated T cells, NK cells, and plasmacytoid dendritic cells (DCs) [41,42]. Notably, CD4-positive/CD25-high/Foxp3-positive/LAG-3-positive T cells were preferentially expanded in patients with cancer [43]. LAG-3 structurally resembles CD4 and binds to major histocompatibility complex (MHC) class II on CD4-positive T cells, resulting in the downregulation of their antigen-mediated activity [44,45,46]. LAG-3 and PD-1 synergistically prevent autoimmunity and promote immune escape in cancer [47,48]. An in vitro experiment reported that T cells co-expressing LAG-3 and PD-1 were frequently seen in bone marrow samples from patients with relapsed AML [49]. An in vivo study of murine chronic lymphocytic leukemia showed that dual inhibition of PD-1 and LAG-3 successfully decreased tumor load [50]. Anti-PD-1 antibodies with or without anti-LAG-3 antibody are now being evaluated in early phase trials for solid and hematologic malignancies [51,52], although these results have not been reported yet.

#### 2.1.4. Leukocyte Surface Antigen CD47

CD47, encoded by the CD47 gene on chromosome 3q13.12, is a transmembrane glycoprotein and a ligand for signal-regulatory proteins (SIRPs). SIRPα is expressed on macrophages, DCs, myeloid cells, neurons, and astrocytes. In terms of macrophages and DCs, SIRPα-CD47 binding inhibits their phagocytic function through the cytoplasmic domain of SIRPα called immunoreceptor tyrosine-based inhibition motifs (ITIMs) by recruiting SH2 domain-containing protein tyrosine phosphatase (SHP)-1 and SHP-2 [53,54]. In other words, CD47-expressing cells are prevented from being engulfed, which have been recognized as the “don’t eat me” signal. Another member of the family, SIRPβ2 is expressed on T cells and NK cells. Unlike SIRPα, binding of SIRPβ2 on T cells to CD47 on antigen-presenting cells results in antigen-specific T cell proliferation and T cell activation, although its affinity for CD47 is weaker than that of SIRPα [55]. Although CD47 is broadly expressed in a variety of normal tissues, it is upregulated in human leukemic cells as well as circulating hematopoietic stem cells (HSCs) [56], and are also related to poor prognoses [57]. A preclinical study in which human tumor cells were co-cultured with SIRPαFc (TTI-621) that binds to CD47 demonstrated the anti-tumor effect of CD47-blockade on various solid and hematologic malignancies [58]. A phase Ib study of anti-CD47 monoclonal (Hu5F9-G4) antibody in combination with rituximab for relapsed or refractory B-cell non-Hodgkin lymphomas showed good tolerability and CR rates of 36% [59], although drug-related anemia occurred as an adverse event [60].

#### 2.1.5. Other Checkpoint Molecules

Similar to the CD47 receptor SIRPα, the B- and T-lymphocyte attenuator (BTLA; also known as CD272) and CD200R (also called OX-2 receptor) have cytoplasmic ITIMs which induce inhibitory signaling in immune effector cells [61,62,63]. CD200 overexpression in AML/MDS cells was associated with higher relapse rates and poor prognoses [64]. Blockade of the BTLA and CD200R/CD200 axis has demonstrated enhanced anti-tumor immunity in preclinical settings [65,66]. Nevertheless, no clinical trials have been conducted to date. B7-homolog 3 (B7-H3; also known as CD276) is one of the B7 family molecules normally expressed on activated APCs and negatively regulates T-cell activation [67]. B7-H3 has been discovered in a variety of solid and hematologic malignancies, and is expected to be a pan-cancer target for IO therapy [68]. A preclinical experiment in which B7-H3-targeted CAR-T cells were co-cultured with 10 samples from AML patients in vitro and administered to human AML-transplanted mice showed significant cytotoxicity in four (40%) patient-derived samples and prolonged survival of the xenograft mice [69]. Representative immune checkpoint molecules are summarized in Figure 1.

### 2.2. Potential Immune Targets on Leukemic Cells

#### 2.2.1. Interleukin-3 Receptor Subunit Alpha (IL-3RA) or CD123

Interleukin-3 receptor subunit alpha (IL-3RA), also known as CD123, contains the alpha-subunit of the receptor for IL-3 and is encoded by the IL3RA gene on the X chromosome. Studies have indicated that IL-3RA is overexpressed in AML cells and other hematologic malignancies but is scant in normal hematopoietic stem cells [70,71,72]. Ehninger et al. reported that 77.9% (232/298) of AML patients were positive for CD123, whereas almost none of the healthy donors were [72]. The presence of CD123 on leukemic stem cells is known to be related to the risk of treatment failure [73]. A study analyzing patient specimens showed that co-expression of CD123, CD25, and CD99 in CD34-positive leukemic cells was frequently observed in AML with FLT3-ITD mutations [74]. Yan and his colleagues demonstrated that the expression of CD123 and CD47 in leukemic stem cells increased in chemo-resistant cell lines compared with chemo-naive ones [75]. Interestingly, romidepsin, an HDAC inhibitor, re-sensitized these resistant cells in vitro. A dual-affinity retargeting (DART) molecule targeting both CD123 and CD3, called MGD006, induced dose-dependent killing of AML cell lines in vitro and in vivo [76]. A preclinical study using AML-transplanted mice showed selective anti-tumor effects of CD123-directed CAR-T on leukemic cells [77]. Similarly, engineered T cells that secrete bispecific CD123/CD3 antibody exhibited anti-leukemic effects in a xenograft mouse model [78]. Monoclonal antibodies and ADCs directing CD123 have also shown efficacy in preclinical studies [79,80,81].

#### 2.2.2. Myeloid Cell Surface Antigen CD33

CD33 is a sialic-acid-binding immunoglobulin-like lectin expressed in monocytic/myeloid lineage cells and is encoded by the CD33 gene on chromosome 19q13.41. CD33 is expressed on myeloid cell lines from the progenitor to well-differentiated cells, including neutrophils, monocytes, and tissue-resident macrophages [82]. CD33 shows positive immunostaining in at least 80–90% of patients with AML, thereby indicating its presence. While CD33 expression levels largely differ between patients, its high expression can be seen in AML with NPM1 mutation [83]. GO, a CD33-targeted ADC, has shown clinical efficacy and has been administered to AML patients in practice (mentioned in later sections). Some studies have suggested that the expression levels of CD33 may be positively related to the anti-leukemic effect of GO treatment [84,85,86]. The first-in-ever CD33/CD3-targeted BiTE, called AMG330, prolonged the survival of human AML-transplanted immunodeficient mice [87] and has been tested in early phase trials (mentioned in a later chapter). A novel bifunctional checkpoint inhibitory T-cell engager (CiTE), a bispecific protein for CD3 and CD33 conjugated with the extracellular domain of PD-1, has been recently developed and was found to improve AML in a murine xenograft model [88].

#### 2.2.3. C-Type Lectin-Like Molecule-1 (CLL-1)

C-type lectin domain family 12 member A (CLEC12A; UniProtKB-Q5QGZ9), also known as c-type lectin-like molecule-1 (CLL-1), is an ITIM-containing inhibitory transmembrane glycoprotein expressed on more than 80% of AML blasts as well as leukemic stem cells. The protein is encoded by the CLEC12A gene on chromosome 12p13.31. Although the function of CLL-1 is not fully understood, its involvement in homeostasis in certain inflammatory situations, such as monosodium urate-induced reaction and collagen antibody-induced arthritis has been suggested [89,90]. A previous study suggested that CLL-1 is selectively present on LSCs but absent on normal HSCs, indicating this as an ideal candidate for immune targets [91]. Hutten et al. showed that CLL-1 is also expressed on myeloid and plasmacytoid DCs, enhancing delivery of tumor antigens into DCs, resulting in efficient antigen presentation on CD8-positive T cells, which was not attenuated by CLL-1-targeted antibodies [92]. Two xenograft models in which patient-derived AML cells were transplanted to cynomolgus monkeys and mice, respectively, showed that CLL-1-targeted ADCs exhibited almost complete depletion of leukemic cells and tumor growth inhibition, respectively [93,94]. Other preclinical data have shown significant anti-leukemic potentials of monoclonal/bispecific antibodies and CAR-Ts targeting CLL-1 [95,96,97,98,99,100,101,102,103,104]. A trispecific killer cell engager (TriKE) targeting CLL-1 on leukemic cells and CD16/IL15 on NK cells increased NK cell proliferation and degranulation in leukemic cells, resulting in death in approximately 15% of the AML cells in vitro, which was comparable to that of CD33-targeted TriKE [105].

#### 2.2.4. Other Candidates of Immune Targets

A proto-oncogene protein c-KIT, also known as CD117, is a type III receptor tyrosine kinase expressed in 80–90% of AML blasts and is related to adverse clinical outcome [106]. A second-generation CAR-T targeting c-Kit demonstrated elimination of more than 90% of CD117-positive AML cells in vitro and almost complete depletion (>98%) of CD117-positive marrow cells in xenograft mice [107]. Like c-Kit, FMS-like tyrosine kinase 3 (FLT3) is a type III receptor tyrosine kinase. FLT3 plays an important role in maintaining the survival of normal HSCs [108] and is also expressed in the vast majority of AML cells along with its recurrent mutations (e.g., internal tandem duplication (ITD) and tyrosine kinase domain mutation (TKD)) [109]. While potent FLT3 inhibitors (e.g., gilteritinib [110], midostaurin [111], and quizartinib [112]) are now available in practice, FLT3-directed novel IO approaches have been investigated to overcome resistance to the inhibitors. A preclinical study of an AML-xenograft mouse model showed that cytokine production and proliferation of FLT3-targeted CAR-T cells were strongly increased by FLT3 upregulation (4- to 13-fold) in leukemic cells after exposure to FLT3 inhibitors, which resulted in a 100% response rate when combined with FLT3-targeted CAR-T and FLT3 inhibitors [113]. In addition, targets of IO therapy do not have to be single. The adapter CAR-T (aCAR-T) is a new concept in which antigen recognition of CAR-T cells is split into multiple agents (e.g., CD32, CD33, CD38, CD123, CD135, CD305, and CLL1 for AML) by linker-label-epitopes (LLEs) that allows qualitative regulation of CAR-T function (for example, recognition of four or more antigens to activate and three or less not to do). Preclinical studies of aCAR-T are ongoing, and the results are awaited [114]. Potential IO targets of AML are schematically displayed in Figure 2.

## 3. Novel IO Therapy in Clinical Trials

### 3.1. Immune Checkpoint Inhibitors

#### 3.1.1. Anti-PD-1/CTLA-4 Antibody

Published trials of ICIs are listed in Table 1. In a phase Ib/II study that recruited 51 patients with AML that failed prior therapy, the combination of nivolumab, an anti-PD-1 antibody, and azacitidine demonstrated 18% CR and 15% hematologic improvement [115]. The median overall survival was 9.3 months, which was favorable to historical data in which patients were treated with azacitidine alone as a salvage therapy. The author also reported a phase II study in which a combination of azacitidine, nivolumab, and ipilimumab (an anti-CTLA-4 antibody) brought about CR and CR with incomplete hematologic recovery (CRi) in 6 of 20 (36%) patients and 58% of 1-year survival rate with severe immune-related adverse effects (irAEs) in 26% of the patients [116]. Farhad et al. reported that the combination of nivolumab and conventional induction chemotherapy (e.g., idarubicin plus cytarabine) was feasible for patients with newly diagnosed AML [117]. CR and CRi were observed in 34 (78%) of 44 patients, including 18 cases in which their minimum residual disease (MRD) was undetectable after completion of the induction therapy. The median overall survival of all patients and that of patients who underwent allogeneic hematopoietic stem cell transplantation (allo-HSCT) were 18.5 and 24 months, respectively. Another anti-PD-1 inhibitor, pembrolizumab, was also evaluated in a small pilot study [118]. Ten patients with relapsed or refractory AML received pembrolizumab along with decitabine every other cycle. CR was observed in one patient (10%) and stable disease (SD) in four patients (40%). The median survival time was 7 months. Ipilimumab showed a response in patients with hematologic malignancies that relapsed after allo-HSCT. In a multicenter phase I study, 28 patients with relapsed hematologic malignancies, including 12 patients with AML and 1 patient with MDS, were enrolled [119]. Among 5 patients (23%) who achieved CR, four had AML and one had MDS. Another open-label phase I study that enrolled 29 patients with solid and hematological malignancies, including two patients with AML, who relapsed after allo-HSCT reported that tumor regression was seen in three patients (10%) but none of them had AML [120]. The efficacy of ipilimumab was also evaluated in MDS resistant to HMAs. In a multi-center phase I study, 29 patients received ipilimumab after HMA failure [121]. One patient (3.4%) achieved marrow CR lasting 3 months, and 7 patients (24%) remained in SD for more than 46 weeks.

#### 3.1.2. Anti-TIM-3 Antibody

TIM-3 blockade can be another strategy for disarming the immune-escaping mechanisms of tumor cells. A phase Ib study testing anti-TIM-3 antibody (MBG453) in combination with decitabine for patients with high-risk MDS and AML reported that 9 of 31 (29%) patients with AML achieved partial response or better response and eight (25.8%) additional patients showed more than a 50% reduction in marrow blasts [122]. Severe (Grade 3–4) irAE, liver toxicity, and arthritis occurred in 7% of the patients; this result was comparable to those of other ICI monotherapy.

### 3.2. CAR-T Therapy and Its Relatives

CAR-T cells are engineered peripheral T cells, often of autologous origin, which have an extracellular antigen-recognition domain, commonly single-chain variable fragments derived from monoclonal antibodies, conjugated with intracellular signal domains (as shown in Figure 2). The first-generation CAR has only one signal domain (CD3 zeta chain) and the other descendants have additional co-stimulatory structures (CD28 or 4-1BB for 2nd generation and CD28 plus 4-1BB or OX40 for 3rd generation). A variety of CAR-Ts are now under development. While CD19-directed CAR-T cell therapy has been successful for B-ALL, the application of the concept for AML has been delayed due to the lack of suitable targetable surface antigens until recently. Ritchie et al. reported a phase I study of the first-in-human and proof-of-concept CAR-T therapy targeting Lewis-Y antigen in patients with relapsed/refractory AML (RR-AML) [123]. Although the clinical outcome was modest, if transient, its feasible transduction efficiency (14–38%) and persistence of CAT-T cells within the body (up to 10 months) as well as its acceptable safety profile were displayed. Among a number of surface antigens, CD33, CD123, and CLL-1 have been eagerly investigated for CAR-T therapy. Wang et al. conducted a clinical trial in which a patient with refractory AML received CD33-directed autologous CAR-T therapy followed by marked reduction of marrow blasts lasting 9 weeks [124]. An early-phase result of CD33-targeted CAR-NK therapy (autologous NK cells with CAR) in patients with RR-AML was reported by Tang and his colleagues [125]. One of three patients who received CAR-NK therapy showed increased levels of serum interleukin (IL)-6 and IL-10 on day 6, followed by a decrease in the MRD level and WT1 copy-numbers, although the clinical benefit was insufficient. Co-administration of engineered DCs with CD33-directed CAR-T cells may enhance anti-leukemic activity through the production of DC-derived IL-12 [126]. A novel compound CAR-T targeting both CD33 and CLL-1 was tested in a phase I study in which a 6 year-old female with Fanconi’s anemia-associated juvenile myelomonocytic leukemia carrying FLT3-ITD mutation that had been heavily treated with multiple therapies, including a FLT3 inhibitor [127]. After CAR-T infusion following lymphodepletion therapy, she achieved CR with negative MRD on day 19, which allowed her to undergo allo-SCT. Sallman et al. reported a phase I study of CAR-T therapy targeting NKG2D, which is expressed on a variety of solid and hematologic tumors, in which three of seven (42%) patients with RR-AML achieved CR [128]. Although only a small number of patients with AML have been involved in clinical trials of CAR-T therapy, the results are promising and further investigations are expected. A summary of early phase trials is shown in Table 2.

### 3.3. Bispecific and Trispecific Antibodies: BiTE and DART

Bispecific antibodies are artificially synthetized small molecules consisting of two different antigen-recognition domains derived from variable regions of monoclonal antibody. Both BiTEs and DARTs recognize CD3 and engage effector T cells with tumor cells. CD33 and CD123 are popular as leukemia-specific targets in recent investigations. BiTEs are composed of a single-chain variable fragment (Fv), and DARTs are made by cross-linking two Fvs [129]. A CD33-targeted BiTE AMG330 was tested in a phase I study of patients with RR-AML [130]. In total, four patients out of 35 (11.4%) participants, including those who received a low dose of AMG330, achieved CR/CRi, and treatment-related severe adverse effects including up to Grade 2 cytokine-releasing syndrome (CRS) were seen in 15 (42.9%) patients. Another CD33-targeted BiTE AMV564 showed reduction of marrow blasts in 12 of 18 (66.7%) patients with RR-AML and up to Grade 2 CRS in one (5.6%) patient [131]. Currently, two agents of CD123-targeted antibodies, XmAb14045 (also known as SQZ622) and flotetuzumab, have been evaluated in early phase studies of patients with RR-AML. A CD33-directed BiTE XmAb14045 achieved CR/CRi in three of 13 (23.1%) participants who received sufficient doses of the agent and up to Grade 3 CRS in 49 of 63 (77.8%) patients who received any dose in the initial stage of the phase I study [132]. Flotetuzumab is a CD123-/CD3-bispecific DART that has undergone a phase I/II study [133]. Of twenty-seven evaluable patients who received its recommended dose, five (18.5%) patients achieved CR/CRi and four (13.3%) patients suffered from Grade 3 or more severe CRS. The agent is also being evaluated in combination with an anti-PD-1 antibody, MGA012, in expectation of more potent clinical benefits [134]. A summary of early phase trials is shown in Table 3.

### 3.4. Antibody-Drug Conjugate

CD33 has been recognized for decades as the most popular immune target of AML. An anti-CD33 ADC, GO, showed CR rates of 63% with 2-year overall survival of 41% in patients with relapsed/refractory CD33-positive AML [8]. A meta-analysis of five randomized trials in which GO was combined with induction chemotherapy in patients with newly diagnosed AML concluded that this combination significantly reduced relapse within 5 years (hazard ratio (HR) 0.81 (0.73–0.90)) and slightly prolonged overall survival (HR 0.90 (0.82–0.98)) without improving response rates (HR 0.91 (0.77–1.07)) [9]. GO monotherapy also showed moderate survival benefit (median survival 4.9 months vs. 3.6 months) without increasing the severe adverse effects compared with best supportive care in the elderly with newly diagnosed AML who were unsuitable for intensive chemotherapy [135]. As mentioned in the earlier sections, novel ADCs targeting CD123 and CLL-1 (e.g., IMGN632 [81] and SL-101 [80] for CD123 and CLT030 [94] for CLL-1) have been shown to have anti-leukemic activity. In the future, they should be evaluated in clinical trials.

## 4. Conclusions

As knowledge of immune inhibitory molecules and leukemic antigens has accumulated, clinical use of IO therapy for AML has come closer to reality. While conventional chemotherapy, with or without SCT, is still pivotal in curative treatment and/or disease control for AML because of its high proliferation intensity, molecular targeted approaches (e.g., GO and FLT3 inhibitors) remain a practical alternative, especially for patients with relapsed/refractory or intolerable AML to intensive treatment. Clinical experience of ICIs in solid tumors has shown that IO therapy brings long-term disease control in at least 10–20% of the patients with metastatic cancers (more favorable in malignant melanoma, renal cell cancer, and non-small cell lung cancer with high PD-L1 expression, and MSI-high tumors) even after discontinuation of chemotherapy. Furthermore, concomitant use of chemotherapy and/or radiation could enhance the efficacy of IO therapy [136,137], although it has not yet been proven in hematologic malignancies. Long-term follow up of CD19-CAR-T therapy (tisa-cel) for relapsed/refractory B-ALL showed more than 5 years of estimated progression-free survival in approximately 40% of patients with low disease burden [6]. Selective eradication of cancer stem cells and/or sustained establishment of anti-tumor immunity might be, at least theoretically, able to bring about a true cure for patients with AML who are relapsed/refractory or ineligible for curative treatment. Further advances and clinical applications of IO therapy for AML are awaited.

## Figures and Tables

**Figure 1 ijms-22-01944-f001:**
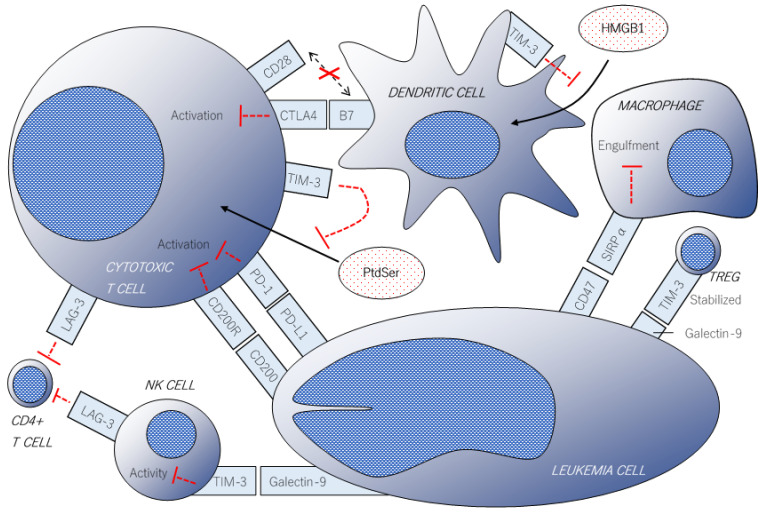
The scheme of immune checkpoint molecules associated with anti-leukemic immunity. Direct cytotoxicity in leukemic cells is attenuated by the binding of programmed cell death-1 (PD-1) on T cells and programmed cell death-ligand 1 (PD-L1) on cancer cells. T-cell activation via B7 family molecules on dendritic cells is canceled by cytotoxic T-lymphocyte-associated protein 4 (CTLA-4). Phagocytic activity of macrophages is dampened by the “don’t eat me” signals of CD47 on leukemic cells. Lymphocyte activation gene-3 protein (LAG-3) expressed on T cells or natural killer cells inhibit the activity of CD4-positive T cells. T-cell immunoglobulin and mucin-domain containing-3 (TIM-3) has multiple inhibitory mechanisms such as attenuation of natural killer (NK) cell activity/stabilization of regulatory T cells by binding to galectin-9 and neutralizing the pro-inflammatory effect of HMGB1 and phosphatidylserine (PtdSer).

**Figure 2 ijms-22-01944-f002:**
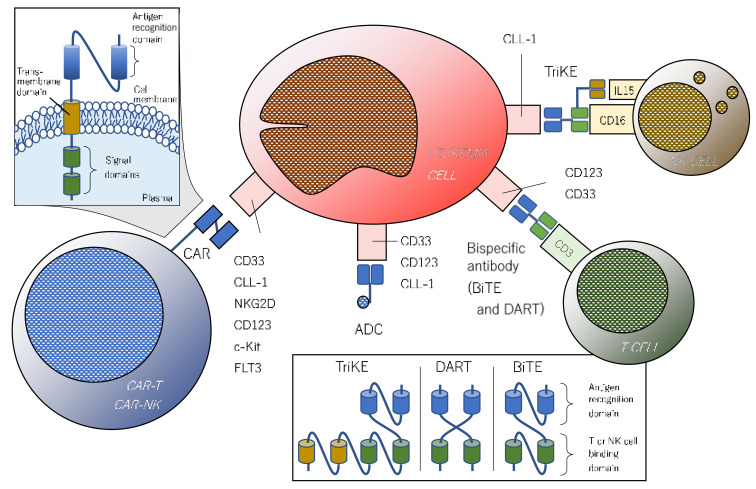
Potential immune targets being investigated in preclinical and early-phase trials. CAR consists of an antigen-recognizing extracellular domain and an intracellular signal domain(s). CAR-T or natural killer-targeting CD33, CLL-1, and pan-cancer antigen NKG2D have been evaluated in phase I trials. While bispecific T-cell engager (BiTE) and trispecific killer cell engager (TriKE) are composed of a single-chain variable fragment (scFv), dual-affinity retargeting (DART) is composed of two cross-linked variable fragments. Bispecific antibodies (BiTEs and DARTs) targeting CD33 and CD123 have been tested in early-phase trials. Gemtuzumab ozogamicin (GO), a CD33-directed antibody-drug conjugate, has already been used in clinical practice.

**Table 1 ijms-22-01944-t001:** A summary of clinical trials of immune checkpoint inhibitors (anti-PD-1, CTLA4, and TIM-3 antibodies) for AML and/or MDS. HL: Hodgkin lymphoma, NHL: non-Hodgkin lymphoma, MM: multiple myeloma, MPN: myeloproliferative neoplasm, ALL: acute lymphoid leukemia, CML: chronic myeloid leukemia, CLL: chronic lymphocytic leukemia, RCC: renal cell carcinoma, allo-HSCT: allogeneic hematopoietic stem cell transplantation, HMA: hypomethylating agent, CR: complete remission, CRi: CR with incomplete hematologic recovery, HI: hematologic improvement, PR: partial remission, SD: stable disease, OS; overall survival.

Author	Object(s)	Disease State	Agent(s)	Dosing	Phase	Response Rate		Median Survival
Daver, et al., 2016	AML	Relapsed after prior therapy	Nivolumab	3 mg/kg on Day 1, 14 (every 4–5 weeks)	Ib/II	CR/CRi HI	18% (6/51) 15% (5/51)	9.3 mo. [1.8–14.3]
			+ Azacitidine	75 mg/m^2^ on Day 1–7 (every 4–5 weeks)				
Daver, et al., 2018	AML	Relapsed or refractory	Nivolumab + Azacitidin	[Cohort 1] Not Reported	II	CR/CRi HI Prolonged SD	21% (15/70) 10% (7/70) 9% (6/70)	16.1 mo.
			Nivolumab+ Azacitidin+ Ipilimimab	[Cohort 2] Not Reported	II	CR/CRi Prolonged SD	36% (6/20) 10% (2/20)	Not Reached (1-yr. OS 58%)
Ravandi, et al., 2019	AML and high-risk MDS	Newly diagnosed	Idarubicin + Cytarabine + Nivolumab	12 mg/m^2^ on Day 1–3 1.5 g/m^2^ on Day 1–4 3 mg/kg (every 2 weeks) *started on Day 24	II	CR/CRi Negative MRD	78% (34/44) 41% (18/34)	18.5 mo. [10.8–28.8]
Davids, et al., 2016	Hematologic malignancies (AML, HL, NHL, MDS, MM, MPN, ALL)	Relapsed after allo-HSCT	Ipilimumab	3 or 10 mg/kg (every 3 weeks for 4 doses then every 12 weeks for upto 6 doses) *All reseposive cases recieved 10 mg/kg.	I/Ib	CR PR	23% (5/28) 9% (2/22)	Not Reported
Bashey, et al., 2009	Malignancies (AML, HL, NHL, MM, CML, CLL, Breast cancer, RCC)	Relapsed after allo-HSCT	Ipilimumab	0.1 to 3.0 mg/kg (every 60 days) *Dose-escalating model.	I	CR PR	6.9% (2/29) 3.4% (1/29)	24.7 mo.
Zeidan, et al., 2018	MDS	Refractory to HMAs	Ipilimumab	3 or 10 mg/kg (every 3 weeks for 4 doses then every 12 weeks for upto 8 doses)	I	Marrow CR Prolonged SD	3.4% (1/29) 24% (7/29)	Not Reported
Lindblad, et al., 2018	AML	Relapsed or refractory	Pembrolizumab + Decitabine	200 mg/body (every 3 weeks) 20 mg/m^2^ on Day 8–12, 15–19 (every 6 weeks)	I/II	CR SD	10% (1/10) 40% (4/10)	7 mo. [2,3,4,5,6,7,8,9,10,11,12,13,14]
Borate, et al., 2019	AML and high-risk MDS	Ineligible to standard therapy	MBG453 (anti-TIM-3) + Decitabine	Escalating dose from 240 to 800 mg/body (every 2 weeks or 4 weeks) 20 mg/m^2^ on Day 1–5 (every 4 weeks)	Ib	CR/CRi PR Blasts halved	23% (7/31) 6% (2/31)2 6% (8/31)	Exposure durations 2.1–17.9 months

**Table 2 ijms-22-01944-t002:** A summary of early phase trials of CAR-T and CAR-NK therapy. JMML: juvenile myelomonocytic leukemia, Rel./ref.: relapsed or refractory, CR: complete remission, CRh: CR with partial hematologic recovery, CRi: CR with incomplete hematologic recovery, MRD: minimal residual disease.

Authors	Objects	Disease State	Agents	Target(s)	Phase	Clinical Outcome
Rithchie, et al., 2013	AML	Rel./ref.	CAR-T (2nd gen.)	Lewis-Y antigen	I	Transient decrease of blasts in 1 of 4 patients 14–38% of Transduction efficiency
Wang, et al., 2015	AML	Rel./ref.	CAR-T (2nd gen.)	CD33	I	Marked reduction of marrow blasts for 9 weeks in 1 patient
Sallman, et al., 2018	Solid tumors and hematologic malignancies	Rel./ref.	CAR-T	NKG2D	I	1 CRh and 2 CRi of 7 patients with AML
Liu, et al., 2018	JMML	Rel./ref.	Compound CAR-T	CLL-1 and CD33	I	CR with negative MRD in 1 patient
Tang, et al., 2018	AML	Rel./ref.	CAR-NK	CD33	I	Decrease of MRD and WT-1 in 1 of 3 patients

**Table 3 ijms-22-01944-t003:** A summary of clinical trials of bispecific antibodies for relapsed/refractory AML (RR-AML). BiTE: bispecific T-cell engager, DART: dual-affinity retargeting molecule, ICI: immune checkpoint inhibitor, CR: complete remission, CRi: CR with incomplete hematologic recovery, CRS: cytokine releasing syndrome.

Author	Object	Agent(s)	Class	Targets	Phase	Clinical Outcome	
Ravandi, et al., 2018	RR-AML	AMG330	BiTE	CD33 and CD3	I	CR/CRi CRS	11.4% (4 of 35) 42.1% (15 of 35)
Eissenberg, et al., 2018	RR-AML	AMV564	BiTE	CD33 and CD3	I	CR/CRi CRS	66.7% (12 of 18) 5.7% (1 of 18)
Ravandi, et al., 2018	RR-AML	XmAb14045	BiTE	CD123 and CD3	I	CR/CRi CRS	23.1% (3 of 13) 77.9% (49 of 63)
Uy, et al., 2018	RR-AML	Flotetuzumab	DART	CD123 and CD3	I/II	CR/CRi Severe CRS	18.5% (5 of 27) 13.4% (4 of 30)
Wei, et al., 2019	RR-AML	Flotetuzumab +MGA012	DART+ ICI	CD123 and CD3 plus PD-1	I	Not reported	

## Data Availability

Not applicable

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
