# Peer review of "Emerging Immunotherapy for Acute Myeloid Leukemia"

_ijms, 2021, doi:10.3390/ijms22041944_

Round 1
Reviewer 1 Report
I have no doubt the science about this manuscript. However the authors should check the grammar before it can be accepted.
Author Response
We reply in the attached file.

Reviewer 2 Report
Please offer the reference for the ICI use in melanoma at the beginning of the introduction. Also, please change "malignant melanoma" to "melanoma".
Is it necessary to offer the uniprot accession numbers for the mentioned proteins, especially considering that this is a review?
Some parts from Figure 2 are not seen so well. Please offer a Figure with a better resolution. Also, please use a wider pallette of colours to distinguish between the cells better in the 2 figures from your manuscript.
Overall, this is a good systematisation of the potential immunotherapies for AML.
Author Response
We reply in the attached file.

Reviewer 3 Report
The manuscript entitled “Emerging Immunotherapy for Acute Myeloid Leukemia” is suitable for publication in ‘International Journal of Molecular Sciences’ after minor revision. In this study authors investigated several immune checkpoint molecules and immune targets in leukemic cells. Recent studies have suggested the potential clinical benefits of immuno-oncology (IO) therapy against acute myeloid leukemia (AML), especially targeting CD33, CD123, and CLL-1, as well as immune checkpoint inhibitors (e.g., anti-PD-1 and anti-CTLA4 antibodies) with or without conventional chemotherapy. Early-phase clinical trials of chimeric antigen receptor (CAR)-T or natural killer (NK) cells for relapsed/refractory AML showed complete remission (CR) or marked reduction of marrow blasts in a few enrolled patients. Bi-/tri-specific antibodies (e.g., BiTE and DART) exhibited 11–67% CR rates with 13–78% risk of cytokine-releasing syndrome (CRS). Conventional chemotherapy in combination with anti-PD-1/anti-CTLA4 antibody for relapsed/refractory AML showed 10–36% CR rates with 7–24 month-long median survival.
Here are the some points below:
- “26” must be in square brackets in “Tregs,26 helps”.
- Comma must be prior to square brackets in “suppressing T cells[27],”.
- Comma must be prior to square brackets in “NK cells[35],”.
- “Was” must be removed in “the expression of CD123 and CD47 in leukemic stem cells was increased”.
- Between must not be divided as be-tween at the end of the sentence.
- Although must not be divided as alt-hough at the end of the sentence.
- Figure 2 must be totally corrected as it is not clear.
- As authors abbreviated gemtuzumab ozogamicin as “GO”, they do not need to use long form in all manuscript.
- Goebeler M-E, Knop S, Viardot A, et al. Bispecific T-Cell Engager (BiTE) Antibody Construct Blinatumomab for the Treatment of Patients With Relapsed/Refractory Non-Hodgkin Lymphoma: Final Results From a Phase I Study. J Clin Oncol. Published online February 16, 2016. doi:10.1200/JCO.2014.59.1586 must be rewritten with its volume, issue and pages.
- Schuster SJ, Bishop MR, Tam CS, et al. Tisagenlecleucel in Adult Relapsed or Refractory Diffuse Large B-Cell Lymphoma. N Engl J Med. Published online December 1, 2018. doi:10.1056/NEJMoa1804980 must be rewritten with its volume, issue and pages.
- Park JH, Rivière I, Gonen M, et al. Long-Term Follow-up of CD19 CAR Therapy in Acute Lymphoblastic Leukemia. N Engl J Med. Published online January 31, 2018. doi:10.1056/NEJMoa1709919 must be rewritten with its volume, issue and pages.
- In reference “Folgiero V, Cifaldi L, Pira GL, Goffredo BM, Vinti L, Locatelli F. TIM-3/Gal-9 interaction induces IFNγ-dependent IDO1 expression in acute myeloid leukemia blast cells. J Hematol OncolJ Hematol Oncol. 2015;8(1):36. doi:10.1186/s13045-015-0134-4” one of J Hematol Oncol must be deleted.
- In reference “Herrmann M, Krupka C, Deiser K, et al. Bifunctional PD-1 x αCD3 x αCD33 fusion protein reverses adaptive immune escape in acute myeloid leukemia. Published online 2018:24.” Journal name, volume, issue and pages must be written.
- In reference “Dagoglu N, Karaman S, Caglar HB, Oral EN. Abscopal Effect of Radiotherapy in the Immunotherapy Era: Systematic Review of Reported Cases. Cureus. 11(2). doi:10.7759/cureus.4103” page number must be added.
Author Response
We reply in the attached file.
